# Similarity of TOPSIS results based on criterion variability: Case study on public economic

Roman Vavrek *, Jiří Bečica

Department of Public Economics, VŠB–Technical University of Ostrava, Ostrava, Czech Republic

* roman.vavrek@vsb.cz

## Abstract

In the real world, acceptance of a decision is conditional on the availability of a great volume of data. Selection of a suitable solution on the basis of this data represents a problem that multi-criterial methods (MCDM) are applied to. The issue of which of these should be favoured during their use involves a specification of the importance of the assessed criteria. The goal of the presented research is to quantify the differences (symmetry) in assessment using selected MCDM methods (Technique for Order of Preference by Similarity to Ideal Solution–TOPSIS), while applying an absolute and relative variability of the assessed criteria to a determination of their importance. The obtained results indicate that the order of the assessed subject (alternative) is not directly influenced by the method of determining the variability of the assessed criteria. We can also state that the degree of concurrence in the order of application of the TOPSIS technique, in combination with both approaches expressed by the Jaccard index, is relatively low.

## Introduction

At a time of increasing global competition, which we do not identify as just the 21$^{st}$ century, it is necessary to pay increasing attention to the effective expenditure of funds [1, 2] or the identification and selection of alternative sources [3]. Decisions on the basis of multiple criteria are gaining in popularity, and the application of this method can be found in various areas of the public and private sectors. The problem that multi-criteria decision-making methods (MCDM) are engaged in resolving is to find and assess the best alternative from the available options [4]. According to [5], these methods represent the process of selecting one alternative from the available options on the basis of a pre-defined set of criteria, which are usually of differing importance. Their purpose is to combine selected criteria and the method of determining their importance into one assessment indicator [6]. This takes into consideration the preferences of the decision-maker, and therefore the actual result, order or recommendation can be modified depending on these preferences [7]. We can encounter a wide range of MCDM methods in practice, as for example PROMETHEE–Preference Ranking Organization Method for Enrichment Evaluations [8], ELECTRE–Elimination and Choice Expressing the Reality [9], TOPSIS–Technique for Order of Preference by Similarity to Ideal Solution [10], VIKOR–Vlse Kriterijumska Optimizacija Kompromisno Resenje [11], IDRA–Intercriteria

**Data Availability Statement:** All data files are available from the Zenodo database (https://doi.org/10.5281/zenodo.6565587).

**Funding:** Funder: VŠB-TU Ostrava Grant no.: SP2022/29 Grant title: Application of selected

mathematical and statistical methods in the conditions of public economy Grant receiver: Roman Vavrek The funders had no role in study design, data collection and analysis, decision to publish, or preparation of the manuscript.

**Competing interests:** The authors have declared that no competing interests exist.

Decision Rule Approach [12], EVAMIX–Evaluation of Mixed Data [13], and others. According to [14], despite the large number of methods, there is no "perfect" method which could be used to resolve all problems.

Consequently, multi-criteria methods can be classified into two main categories: the discrete MCDM or MADM (Multi-Attribute Decision-Making) and continuous multi-objective, decision-making (MODM) optimization methods [15]. In a more complex manner, we can find a division into 4 groups [16, 17]. One of the simplest taxonomies including the TOPSIS technique is offered by [18], see the following Fig 1:

The presented research focuses on evaluation of 72 subjects (district towns) using 6 unique criteria. The TOPSIS technique, which is an appropriate tool for decision-making on the basis of incomplete data, according to [19], is therefore used for this purpose. According to [20, 21], the scope of the chosen data is not determinative for use of this method, i.e., data of any scope can be used. This method proposes the minimization of the distance with respect to the ideal and, simultaneously, the maximization of the distance with respect to the anti-ideal [22]. The strong points of this method include the directness and simplicity of calculations [22, 23], and the ability to work with all types of criteria [21] and complexity [24]. [20, 21] consider the absence of the option to allocate weights to the monitored criteria and the absence of consistent supervision by the decision-maker to be the main disadvantages of the TOPSIS method. For this reason, this method depends on the process of establishing the importance of various attributes with regard to achieving the specified goal, which is also the area of interest of the presented research (as it is possible to choose methods according to the purpose of one's own research).

The aim of this paper is to quantify the differences (symmetry) in assessment using the TOPSIS technique while using an absolute and relative variability of the assessed criteria to a determination of their importance. The selected Coefficient of Variance (CV) and Standard Deviation (SD) methods were chosen for this purpose, i.e., a relative and absolute assessment of the variability of the assessed criteria. Both represent moment characteristics of variability, but their calculation itself is different. The actual case study is preceded by an execution of the theoretical foundation of the MCDM issue, focusing on the TOPSIS technique and the methods of determining the importance of the assessed criteria. The research sample, criteria from the aspect of public economics, is described within the terms of the methodology. Attention is also focused on determining importance, as determined by means of the two aforementioned methods and the mechanism of the mathematical-statistical methods used. In the conclusion, the obtained results are summarised and potential future research is simultaneously outlined.

## The TOPSIS technique as a multi-criteria assessment tool

TOPSIS is one of the basic methods for multi-criteria decision-making, and its primary application is for resolving various types of decision-making issues. This method is one of the most frequently used methods, according to [25, 26], who also cite the above-mentioned AHP, ANP or PROMETHEE methods as an alternative. A summary of its application is described, for example, by [27], who registered the annually increasing number of incidents of research/articles in which the application of TOPSIS and other techniques can be found. Selection of the TOPSIS method for the purposes of our research was based on its previous successful use in resolving decision-making problems of a similar nature. Its application can be found in the environmental [28–30], transport [31, 32], logistical [33] and many other fields (see [34, 35]).

[27] expresses the principle of the TOPSIS technique using Fig 2, to which [36] also refers during a description of this method. Each white ball represents a concrete alternative, i.e., one assessed subject. The grey ball represents a Negative Ideal Solution (NIS) alternative, i.e., a real

| MCDM methods | | |
|---|---|---|
| **ranking methods** | TOPSIS, Lexicographic | |
| **scoring methods** | Weighted Average, Weighted Product | |
| **non-compensatory methods** | Dominance, Maxmin, Maxmax | |
| **comparative methods** | AHP, ELECTRE | |

**Fig 1. Taxonomy of the TOPSIS technique.**

or hypothetical alternative (subject) with the worst values of the individual criteria. The black ball represents a Positive Ideal Solution (PIS) alternative, i.e., a real or hypothetical alternative (subject) with the best values of the individual criteria. The best rated is the alternative (one of the white balls) which is the furthest away from the grey ball (NIS) and also the closest to the black ball (PIS). For example, [37–39] discuss an actual calculation of the TOPSIS technique, during which time the individual steps are as follows:

1. In the first step, one needs to create a basis for the decision-making matrix, consisting of individually assessed subjects (alternatives) and previously defined criteria:

$$D = \begin{pmatrix}
 & X_1 & X_2 \ldots & X_j \ldots & X_n \\
A_1 & x_{11} & x_{12} \ldots & x_{1j} \ldots & x_{1n} \\
A_2 & x_{21} & x_{22} \ldots & x_{2j} \ldots & x_{2n} \\
\vdots & \vdots & \vdots & \vdots & \vdots \\
A_i & x_{i1} & x_{i2} \ldots & x_{ij} \ldots & x_{in} \\
\vdots & \vdots & \vdots & \vdots & \vdots \\
A_m & x_{m1} & x_{m2} \ldots & x_{mj} \ldots & x_{mn}
\end{pmatrix} \tag{1}$$

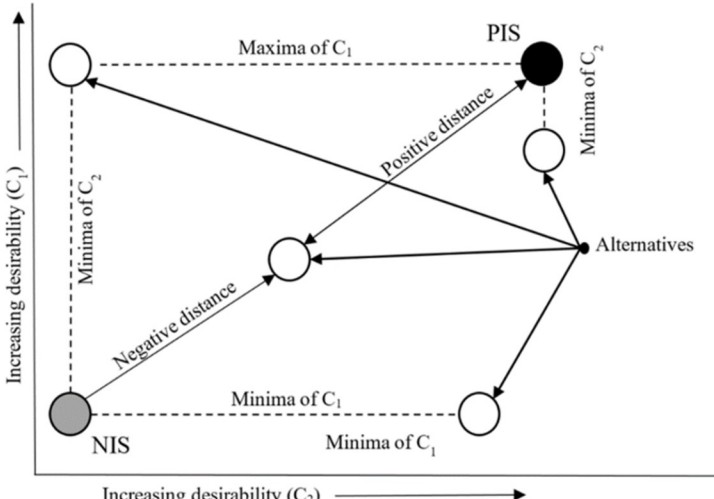

**Fig 2. Graphical presentation of the TOPSIS technique.**

where: Ai = i(th) alternative

$x_{ij}$ = value of the j(th) criterion reached by the i(th) alternative

2. In practice, the assessed criteria may have various parameters (expressed by the various level-of-moment characteristics), for the assessment of which the matrix is standardised in the subsequent step:

$$r_{ij} = x_{ij} / \sqrt{\sum_{j=1}^{j} x_{ij}^2} \qquad (2)$$

where: $r_{ij}$ = normalized value of the j(th) criterion

$x_{ij}$ = value of the j(th) criterion reached by the i(th) alternative

3. The decision-maker's preferences are taken into consideration in the third step of the calculation, which represents the application of the results of the selected method for determining the importance of the assessed criteria:

$$v_{ij} = w_{ij} * r_{ij} \qquad (3)$$

where: $v_{ij}$ = weighted normalized value

$w_{ij}$ = criterion weight

$r_{ij}$ = normalized value

4. Given the fact that the TOPSIS technique works with Euclidean distance, the Positive Ideal Solution (PIS) and Negative Ideal Solution (NIS) must be identified:

$$H_j = \max\left(v_{ij}\right), D_j = \min\left(v_{ij}\right) \qquad (4)$$

where: $H_j$ = PIS

$D_j$ = NIS

5. After identification of these two alternatives, the Euclidean distance between each real alternative (subject) and the mentioned PIS and NIS alternatives, is calculated:

$$d_i^+ = \sqrt{\sum_{j=1}^{k} \left(w_{ij} - H_j\right)^2}, \ d_i^- = \sqrt{\sum_{j=1}^{k} \left(w_{ij} - D_j\right)^2} \qquad (5)$$

where: $d^+$ = distance from PIS

$d^-$ = distance from NIS

6. The resulting assessment parameter is the relative distance to the PIS alternative, by means of which the subjects are assessed:

$$c_i = \frac{d_i^-}{d_i^- + d_i^+} \qquad (6)$$

where: $c_i$ = relative distance from PIS

Within the terms of each MCDM method, one of the most important steps in the calculation is to determine the importance of the assessed criteria; this process takes place in the third step of the calculation for the TOPSIS technique. The approach to resolving this issue is discussed in the following section.

### Variability as a factor for determining the importance of criteria

In general, it can be stated that the result of applying the MCDM method is directly determined by the decision-maker and also by, among other things, which approach or method he selects to determine the importance of individual criteria. One of the classifications of these methods is presented by [40], who identifies 2 groups of methods. [39] works with 3 groups and [41] uses 4 groups of approaches for classifying weights into 4 groups, which represents an expansion of the preceding classifications, during which time the groups in question are as follows:

1. subjective,

2. expert,

3. objective,

4. integrated.

Subjective methods reflect the personality of the decision-maker and his individual preferences (the weight of the indicator is determined on the basis of subjective opinion). Expert assessment is executed by a smaller number of experts in the specific field, during which time the previous application is described by [42–44]. The Fuller method, or Fuller triangle, is the most commonly used of these groups of methods. The third group, i.e., the group of objective methods, assigns a weight to individual criteria on the basis of a previously determined mathematical model unique to each method. This means that the decision-maker does not have direct influence on determining the importance of criteria; a selection is made depending on his preference of the properties of the used data, e.g., according to variability or relations between criteria. The last group is integrated methods, which represent a combination of the approaches described above. The group of objective methods (third group) can include methods such as the Mean Weight method [45, 46], Standard Deviation method [47], Mahalanobis-Taguchi System method [48], λ bi-capacity model [49], Coefficient of Variance method [30], and many others (see [50, 51]). Variability can also be found in research in other forms, e.g., as a moment characteristic [52] or control CV charts [53, 54], or as a method for determining the importance of the assessed criteria [55]. We work with the following two methods for the purposes of this research:

- method utilising relative variability—the Coefficient of Variance method (CV):

$$w_j = \frac{CV_j}{\sum_{j=1}^{n} CV_j} \tag{7}$$

where: $CV_j$ = coefficient of variance of the j(th) criterion

- method utilising absolute variability—the Standard Deviation method (SD):

$$w_j = \frac{SD_j}{\sum_{j=1}^{n} SD_j} \tag{8}$$

where: $SD_j$ = standard deviation of the j(th) criterion

### Research methodology

The purpose of the presented paper is to quantify the differences (symmetry) in assessment using the TOPSIS technique, while applying an absolute and relative variability of the assessed criteria to a determination of their importance. The fulfilment of this purpose should lead to

the solution of the problem posed by the absence of a comparison of this type (with a finite number of alternatives, criteria and external determination of criteria importance using selected methods).

The case study is realised within the environment of public economics, in the context of which the research sample and monitored criteria are described. The following section represents a description of the research sample and criteria and an assessment of the importance of selected criteria, utilising two methods for determining their importance, namely the CV and SD methods. The mechanisms of the mathematical-statistical methods used are described in the last sub-section.

## Research sample and criteria from the aspect of the public economy

The research sample for the presented research is a sample of all district towns in the Czech Republic (72 subjects), which are reported on an international scale within the terms of statistical surveys as natural centres of the LAU1 statistical unit (district centre). The specified statistical units originated within the territory of what is now the Czech Republic in the time of the Austro-Hungarian Empire, under the rule of the Habsburg Monarchy, after acceptance of the so-called Kroměříž Constitution in 1848, and their number was gradually modified in the following decades. Within the terms of analysis, the cost aspect of the sector classification of the budgets of a total of 72 district towns, which forms the full list of towns that are the seats of a district in the Czech Republic, is examined. Although a total of 77 districts (LAU1) are registered in the Czech Republic in relation to official statistics, in practice, the administrative territory of a total of 5 districts (Prague-East, Prague-West, Brno–Country, Plzeň –South and Plzeň-North) is actually the relevant territory of one of three major cities in the Czech Republic (Prague, Brno and Plzeň), and these cities are actually the seats of a multiple number of districts. The areas of the districts remained valid in the Czech Republic even after the extensive reforms to the public administration system in 2003, when their actual budgets (the budgets of district authorities) were abolished, but these administrative centres remained valid for the requirements of courts of law, the police, archives, employment offices, etc. At present, they are primarily utilised as a basic statistical unit in a territory, while the official district code book is created and maintained by the Czech Statistical Office.

The district is a part of the public administration system and can be viewed from various angles. Understanding the entire scope and importance of the function of self-governing municipalities, and subsequently the entire public administration system within a defined territory (state, region, municipality), requires a multi-disciplinary approach [56, 57]. This is applied by administrative science [58], which also defines the approaches to the provision of public services, which are defined by content, tasks and specific goals, and which usually differ between individual public administration units according to the defined area and actual scope of provision [59]. On the basis of the Czech constitution, a municipality is considered to be a basic territorial self-governing community of citizens, forming a territorial unit, and is defined by a border or the cadastral area of the municipality. According to Act No. 128/2000 Sb., on municipalities, each municipality in the Czech Republic should assure the universal development of its territory and meet the needs of its citizens. These needs include public services, which are provided to the residents entirely, or for partial payment, are based on acknowledged public interests, and may differ depending on the defined territory (state, region, municipality) of their provision [60, 61]. In practice, the specific needs of citizens are never specified in detail, and the management of the municipality makes decisions at its deliberation. This results in situations where municipalities of the same size report different results to each other, even on a longer time scale. Differences may be evident between municipalities from the

income aspect [62] of the budget (typically on the level of non-tax and capital revenue) [63], particularly in relation to individual cost categories [64]. Income and costs are individual in the budgets of municipalities, but the contents are standardised according to the valid budget composition, which is specified by Decree of the Ministry of Finance of the Czech Republic No. 323/2002 Sb. Act No. 128/2000 Sb., on municipalities, which also stipulates that during execution of their tasks, municipalities must endeavour to protect the public interest. The valid legislation (particularly the constitution of the Czech Republic) indicates that the municipality is a public corporation with its own budget and property, which it disposes of on the basis of adherence to specific laws for decisions made by elected members of the municipal council. Municipalities in the Czech Republic enter into legal relations in their own name and bear the responsibilities arising from these relations. Each part of the territory of the Czech Republic is part of the territory of one of a total of 6,258 self-governing municipalities, which are typical in that they have one or more cadastral areas and a council elected by the citizens of this municipality.

The foundations of the Czech municipal system are based on operation within a so-called mixed system, during which the municipalities, as local self-governing units, have their own authority (self-government) and transferred authority. The state may only intervene in independent (self-government) authority if required by law, and only in a manner stipulated by the law. The advantage of independent authority is that municipalities are able to cooperate when conducting activities. The transferred authority of municipalities is essentially de-concentrated state administration entrusted to municipalities by laws on the basis of the provisions of Article 105 of the Czech constitution, which states that execution of state administration can be entrusted to self-governing bodies only if stipulated by the law. This therefore concerns indirect state administration, in which the municipality appears similarly to purely executive state administration bodies, and the realised activity is of a sub-legal, executive and ordered nature. In relation to transferred authority, municipalities in the Czech Republic follow valid laws, and the result is state assistance during execution of matters within its jurisdiction and powers, during which time they must follow not only the legal regulations to the degree stipulated by the law, but also government resolutions and guidelines from central administrative authorities. With regard to the mixed system for local public administration, it can be stated that the complexity of the existing system of organisation of territorial public administration in the Czech Republic is also substantially increased by the common regulations for individual areas, in relation to both independent and transferred authority. From the viewpoint of the ordinary citizen, independent and transferred authority subsequently intermingles and overlaps in municipalities. Within the terms of this paper, changes to the cost structure in individual budgets during the period from 2010–2020 for all district towns in the Czech Republic, calculated per permanently residing citizen in the relevant year and registered in the specific territory, are assessed. Classification of the budget components by sector complexly covers all the executed expenses of the selected sample of municipalities. Expenses are classified by individual common attribute into sectors (agriculture, industry, services for residents, social matters, safety and public administration) according to the sector classification of the budget, which is given by the valid decree of the Ministry of Finance of the Czech Republic on budget composition.

## Assessed criteria from the aspect of the public economy

Within the terms of assessment of the budgets of the above-mentioned district towns, the authors of this paper selected individual categories of costs depending on sector classification of the budget composition, which is binding for all municipalities in the Czech Republic on the basis of the valid legislation, specifically:

- R1 – Agriculture, forestry and fishing (in CZK per capita),

- R2 – Industrial and other economic sectors (in CZK per capita),

- R3 – Services for residents (in CZK per capita),

- R4 – Social matters and employment policy (in CZK per capita),

- R5 – State security and legal protection (in CZK per capita),

- R6 – General public administration and services (in CZK per capita).

While the scope of a great number of activities under independent authority depends on decisions made by the municipality and on local conditions, the scope of their execution is not clearly determined in the case of transferred authority. In relation to sector classification of the budget composition (R1 –R6), this concerns expenses for public administration, in which we also classify administration regarding the hearing of offences against public order in state administration and in territorial self-government, offences against public order, offences against property, and offences against co-existence between citizens (unless these were committed by the breach of special legal regulations on road traffic), as well as offences in the sphere of the seeking out, protection, use and further development of natural therapeutic resources, including sources of natural mineral water, and spa locations. The municipal authorities of municipalities with expanded authority, which also includes the district towns assessed below, hear offences in matters they administer, along with other offences, if other administrative bodies do not have the authority to hear them.

How accurately it is possible to correctly determine the distribution of a municipality's expenses throughout individual sectors is the subject of assessment, and the criteria for the requirements of multi-criteria assessment are therefore not of an absolute or relative value. Year-on-year changes, or more precisely progress, which is calculated as follows, is assessed.

## The importance of the assessed criteria from the aspect of their variability

As specified above, one approach to determining the importance of criteria is based on the variability of the data/criteria. For the requirements of the presented research, relative variability is recorded using the CV (Coefficient of Variance) method, and absolute variability is recorded using the SD (Standard Deviation) method. The method of quantifying variability was expressed in the differences between the allocated weights (importance) recorded in Fig 3.

There are no statistically significant differences in importance from the aspect of individual criteria (during assessment of the distribution function, mean value–median and standard deviation). The exception to this rule is the third criterion (R3), where the obtained sets of weights have a different distribution function (K-$S_{R3}$ = 0.8; $p < 0.01$) and mean value ($W_{R3}$ = 12; $p < 0.01$). With the exception of the period of 2015/2016, the standard deviation is homogeneous ($LE_{EP6}$ = 5.3405; $p < 0.05$). Differences in the distribution function across individual years are not statistically significant, and the same applies in the case of mean values–medians. However, differences can be observed in selected moment characteristics, e.g., in the variation range or minimum and maximum values, see Table 1.

The mean value of the allocated weights is mostly higher in the case of the measurement of relative variability, i.e., using the CV method. Using this method, the lowest importance of 4 out of 6 monitored criteria (R2 –R5) is higher, during which time in absolute results we can observe a higher importance during use of the SD method. The assessment of the variation range, which is higher for all criteria, is clearly the result of the use of the SD method.

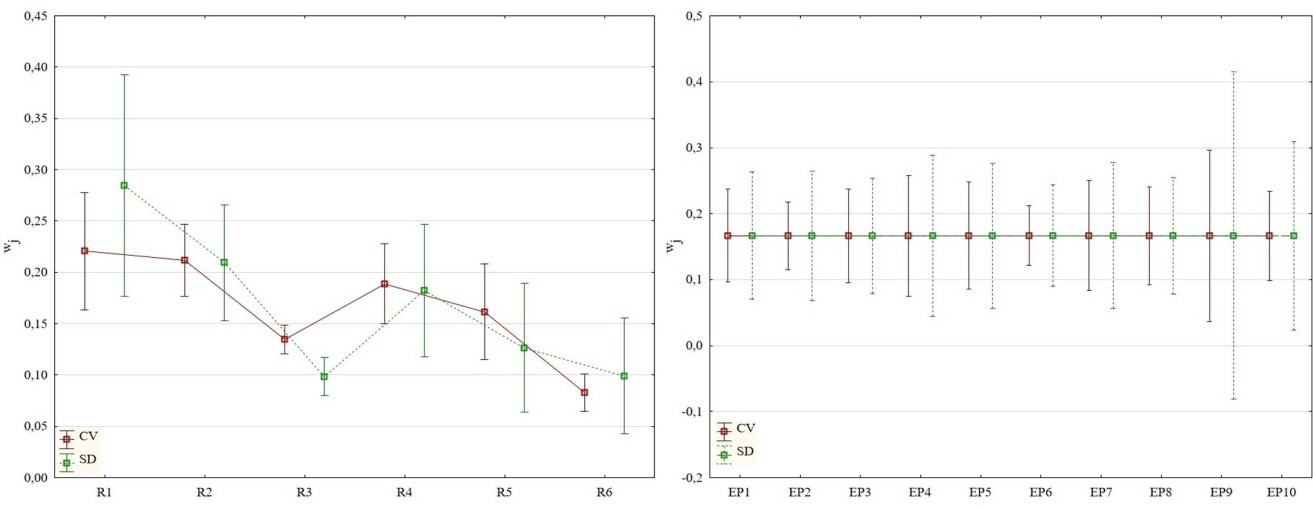

**Fig 3. Average importance of the assessed criteria (whisker: max—min).**

The basic prerequisite for quantification of the results of variability measurement on the results of multi-criteria assessment is therefore partially proven in the heterogeneous determination of the importance of individual criteria. The result of these differences is quantified in the analytical section, i.e., the third chapter.

## Mathematical-statistical methods

The basic method used in the executed analysis is the TOPSIS technique, the calculation of which is described in detail in one of the previous sections. The results, in combination with the CV and SD methods, are subsequently verified and interpreted using selected moment characteristics and the following methods:

• Levene test

$$W = \frac{(N-k)}{(k-1)} \frac{\sum_{i=1}^{k} N_i (Z_i - Z_{..})^2}{\sum_{i=1}^{k} \sum_{j=1}^{N_i} \left( Z_{ij} - Z_{i.} \right)^2} \tag{9}$$

**Table 1. Comparison of selected moment characteristics of criteria weights (CV vs. SD).**

|  |  | R1 | R2 | R3 | R4 | R5 | R6 |
|---|---|---|---|---|---|---|---|
| **average** | CV | 0,220573 | 0,211509 | 0,134732 | 0,188743 | 0,161624 | 0,082819 |
|  | SD | 0,284660 | 0,209309 | 0,098305 | 0,182215 | 0,126529 | 0,098981 |
| **median** | CV | 0,198364 | 0,20174 | 0,130307 | 0,209422 | 0,158018 | 0,080635 |
|  | SD | 0,239495 | 0,203794 | 0,097492 | 0,208699 | 0,116542 | 0,07976 |
| **minimum** | CV | 0,139842 | 0,152941 | 0,107906 | 0,08329 | 0,092666 | 0,032982 |
|  | SD | 0,155956 | 0,104342 | 0,066372 | 0,054085 | 0,045832 | 0,037942 |
| **maximum** | CV | 0,406623 | 0,310420 | 0,171810 | 0,243068 | 0,321600 | 0,115446 |
|  | SD | 0,646662 | 0,299075 | 0,154384 | 0,314219 | 0,348344 | 0,313807 |
| **range** | CV | 0,266781 | 0,157480 | 0,063904 | 0,159778 | 0,228935 | 0,082464 |
|  | SD | 0,490707 | 0,194733 | 0,088012 | 0,260135 | 0,302512 | 0,275865 |

where: k = number of values of the observed categorical variable

N = number of observations

$N_i$ = number of observations in the i(th) group

$Y_{ij}$ = measured value of the j(th) unit of the i(th) group

$\bar{Y}_i$ = average value of the i(th) group

$\tilde{Y}_i$ = median of the i(th) group

$Z_{..}$ = average of groups $Z_{ij}$

$Z_{i.}$ = average of $Z_{ij}$ for the i(th) group

- Kruskal-Wallis test

$$Q = \frac{12}{n(n-1)} \sum_{i=1}^{I} \frac{T_i^2}{n_i} - 3(n+1) \tag{10}$$

where: n = number of observations and sample size

$n_i$ = number of observations in the i(th) group

$T_i^2$ = total number of orders in the i(th) group

- Jaccard index

$$J = \frac{A \cap B}{A \cup B} \tag{11}$$

where: A = set 1 (list of subjects)

B = set 2 (list of subjects)

The first two tests represent commonly used non-parametric tests for testing homoscedasticity, or agreement between the mean values. The Jaccard index is one of the most frequently used indices [65–67] and is used to assess the order of the results of CV-TOPSIS and SD-TOPSIS. All analyses are executed in MS Excel, Statistica 13.4 and Statgraphics XVIII software.

## Results

In the preceding section, differences in importance determined on the basis of the CV and SD methods, i.e., on the basis of relative and absolute assessment of the variability of monitored criteria, among other differences, were established. In this section we gradually present the results of the multi-criteria assessment using the TOPSIS technique (in combination with both the aforementioned approaches), while focusing on quantification of the established differences (see Fig 4).

Results from application of the CV-TOPSIS method show significant differences across the entire assessed period (see Fig 4). Subjects with considerably better assessment, i.e., outliers/extreme observations, can be found in each of the evaluated periods (EP). This fact became evident by rejection of the homoscedasticity of these results (LE = 2.0335; $p < 0.05$) and also by confirmation of the differences between mean values (Q = 350.115; $p < 0.01$). At the same time, on each occasion the results are positively skewed (min. $\gamma_{1(EP8)} = 1.062$), and we therefore observe that a majority of the subjects have an above-average assessment.

Assessment based on the SD-TOPSIS method shows significant differences, which can be indicated on the basis of Fig 5. Statistically significant results became apparent from the aspect of the assessment of homoscedasticity (LE = 2.366; $p < 0.05$) and also during a comparison of mean values (Q = 492.664; $p < 0.01$). During each of the evaluated periods (EP), we observe multiple outliers/extreme observations, along with a high concentration of the majority around the expected value (max. $\gamma_{2(EP5)} = 44.087$).

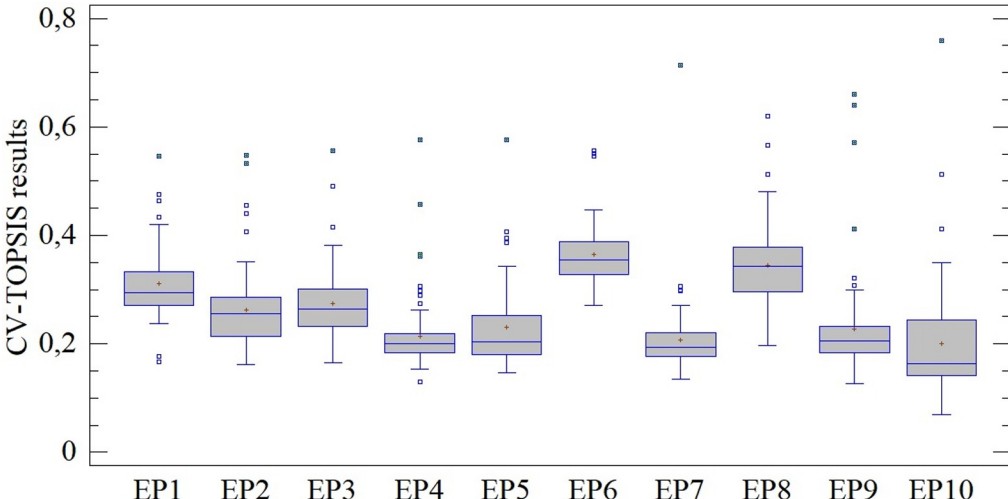

**Fig 4. Results of assessment using the CV-TOPSIS method.**

## Identification of differences in the obtained results

The results of the multi-criteria assessment outlined above change substantially during the monitored period of 2010–2020. Although differences can be observed from multiple angles (e.g., through individual moment characteristics), this does not automatically mean confirmation of the differences between the results of the applied CV and SD methods. Assessment of these differences is carried out on the basis of absolute values (Fig 6), the order of individual subjects (Fig 7), and also the assessment of both aspects simultaneously (Fig 8). In the last step, agreement on the level of individual subjects is monitored by application of the Jaccard index (Fig 9).

When monitoring absolute differences, we can confirm the dominance of the results of applying the CV method (see Fig 6). The results acquired in this manner are mostly higher/better, with the exception of EP5, which is given by the great evenness of the results of the SD-TOPSIS method in this period. During most of the evaluated period (EP), these results are

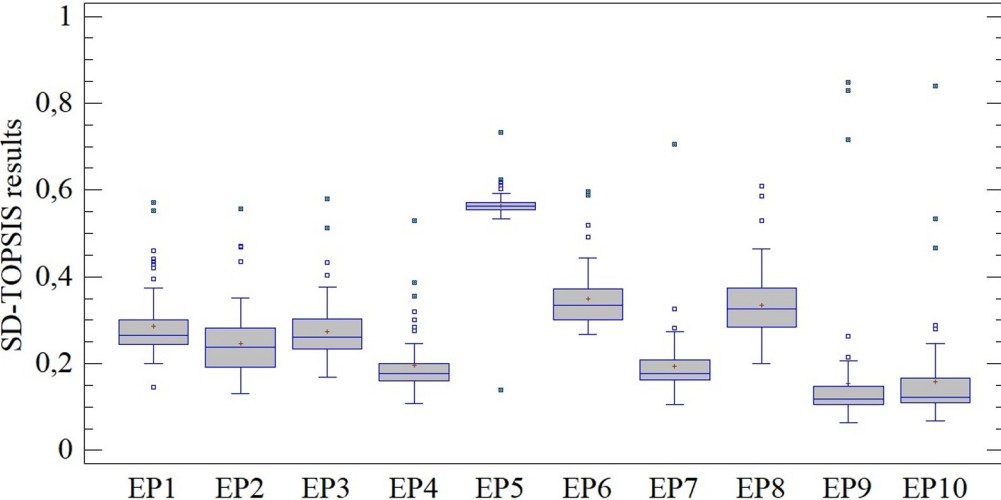

**Fig 5. Results of assessment using the SD-TOPSIS method.**

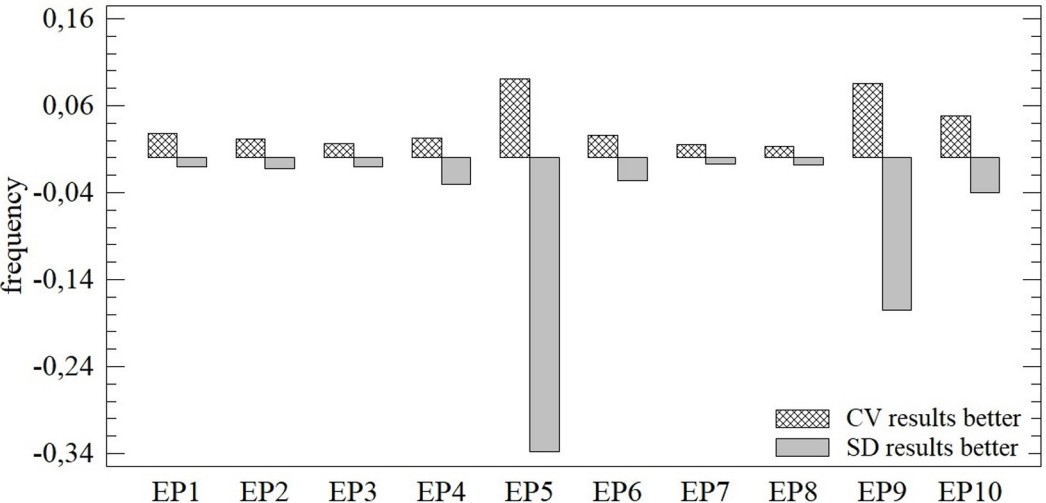

**Fig 6. Average differences in the absolute results of the TOPSIS technique (CV vs. SD).**

better by a 0.129 to 0.90 relative distance to the PIS alternative ($c_i$). In the case of better results using the SD method, their frequency is lower, but is also linked to a greater difference.

The differences arising from the use of various methodologies must also be considered in the context of the results of other assessed subjects, which is portrayed in Fig 7. In 44.44% of the cases for the entire monitored period, the placement of the assessed subject on the basis of the CV-TOPSIS method is better. Better results in relation to application of the other method (SD-TOPSIS) can be observed in 42.50% of cases, i.e., the order did not change in just 13.06% of the cases. From this aspect, it is impossible to identify one of these methods as dominant.

As summarised by Fig 8, in the majority of cases we observe better results if relative variability is used (CV method) to determine the importance of the assessed criteria. Paradoxically, these better results are linked to a decline in overall order (50% of cases), which we credit mainly to the specific situation during the EP5 assessed period.

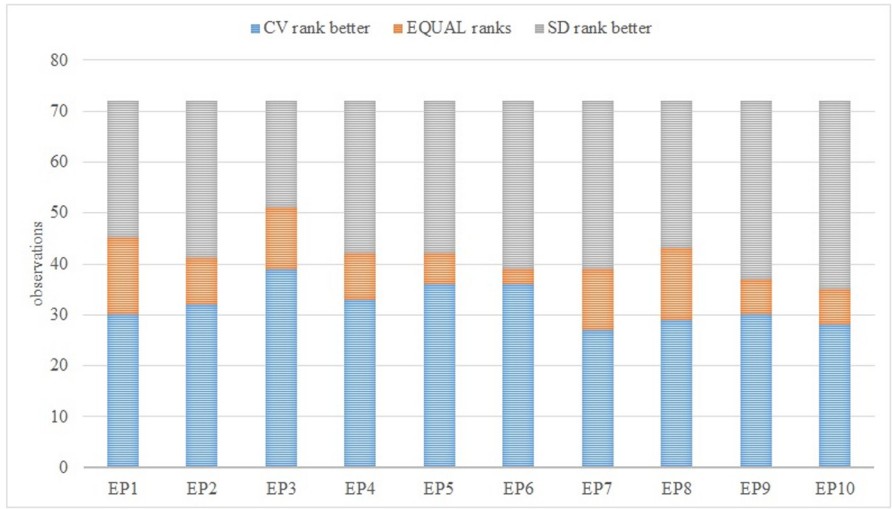

**Fig 7. Comparison of the order of individual subjects on the basis of the results of the TOPSIS method (CV vs. SD).**

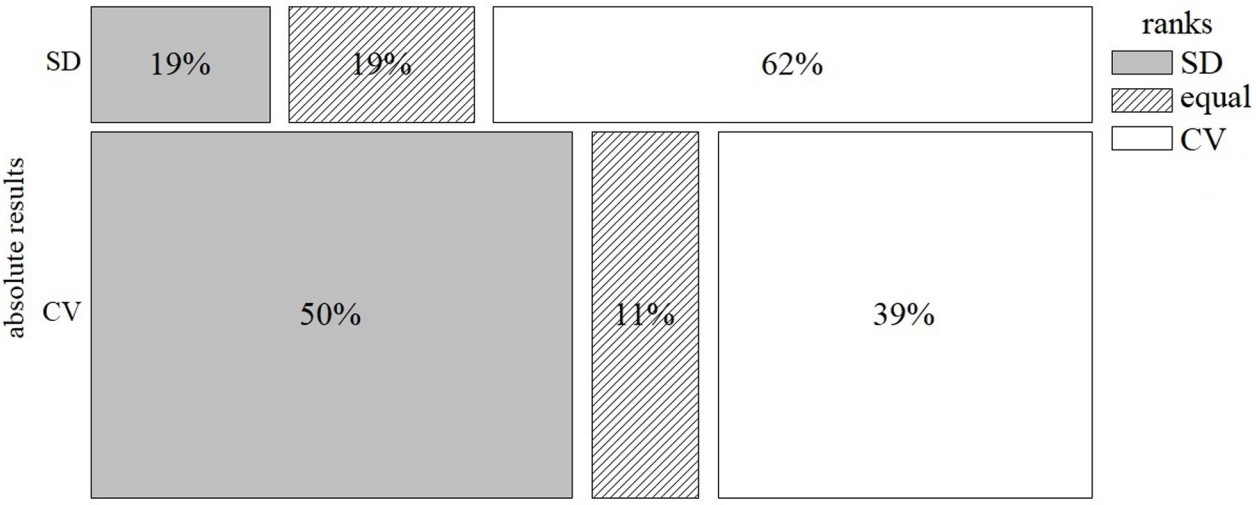

**Fig 8. Structure of individual subjects on the basis of the results of the TOPSIS method (CV vs. SD).**

From the aspect of individual subjects and their placement, we can state that the degree of agreement in their order on the basis of assessment using both methods is relatively low. The position of individual subjects is significantly heterogeneous during the individually assessed periods, when the degree of agreement quantified using the Jaccard index ranged between 4.17% and 20.83%. With regard to the methodology of this index, the "strength" of the heterogeneity determined in this manner cannot be interpreted in more detail (because deviation by one place in the overall order is sufficient).

## Conclusions

Making decisions on the basis of multiple criteria is currently a frequently used approach that can be broadly employed in practice. Within the terms of execution of this method, adequate attention must be devoted to determining the importance of the assessed criteria; this can be approached in different ways. The submitted research presents the application of two methods

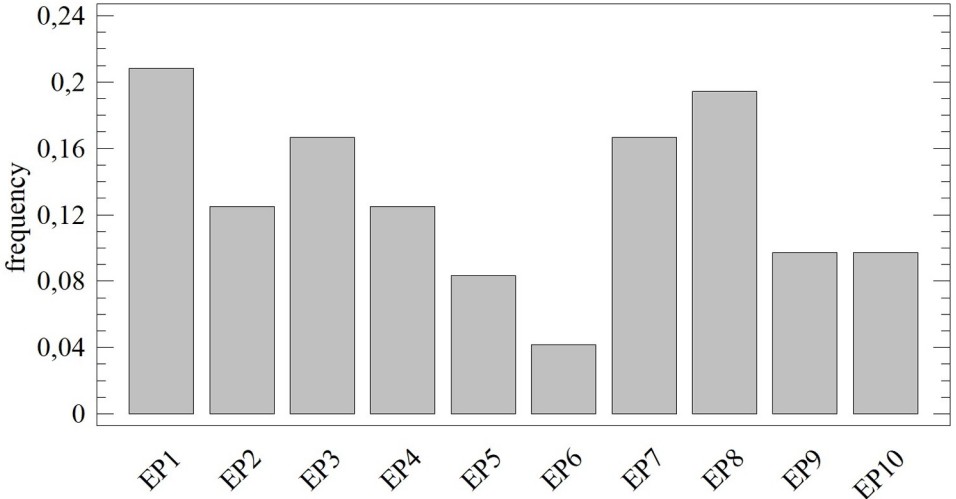

**Fig 9. Comparison of the agreement of the results of the TOPSIS method using the Jaccard index (CV vs. SD).**

for determining the importance of criteria, which are based on variability expressed as relative (CV method) or absolute (SD method), in relation to the TOPSIS method. The described contents of expenses by sector classification of budget elements clearly indicates that these cover the entire range of expenses of all municipalities within the terms of the budget. However, specific expenses may differ between municipalities on a year-on-year basis, which is caused by the different requirements for financing or ensuring specific public services for the residents in the specific territory. Application of the above-mentioned methods resulted in the following findings from a methodological aspect:

- during application of the SD method, higher absolute differences in determined importance were measured in the case of all the monitored criteria,

- during the use of relative variability to determine the importance of the assessed criteria (CV method), the subjects are usually rated more highly,

- the order of the assessed subject is not directly influenced by selection of the CV or SD method,

- the degree of similarity expressed by the Jaccard index is relatively low.

In general, it is necessary to respect the differences arising from both perspectives when determining the importance of the evaluated criteria (absolute and relative). The absolute view cannot be replaced by a relative one, or vice versa. Methods based on the relative importance of the criteria may indicate a better evaluation of the subjects, but without affecting their overall ranking.

In the future, the plan is to follow up on the obtained results by verification using other MCDM methods, such as VIKOR or PROMETHEE, and to verify these conclusions.

## Author Contributions

**Conceptualization:** Roman Vavrek.

**Data curation:** Roman Vavrek.

**Formal analysis:** Roman Vavrek.

**Funding acquisition:** Roman Vavrek.

**Investigation:** Roman Vavrek.

**Methodology:** Roman Vavrek.

**Project administration:** Roman Vavrek.

**Resources:** Roman Vavrek.

**Software:** Roman Vavrek.

**Supervision:** Roman Vavrek.

**Validation:** Roman Vavrek.

**Visualization:** Roman Vavrek.

**Writing – original draft:** Roman Vavrek, Jiří Bečica.

**Writing – review & editing:** Roman Vavrek.

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
