## [Decision Letter · Decision Letter 0]

1 Mar 2022

PONE-D-21-35126Similarity of TOPSIS Results based on Criterion Variability: Case Study on Public EconomicPLOS ONE

Dear Dr. Vavrek,

Thank you for submitting your manuscript to PLOS ONE. After careful consideration, we feel that it has merit but does not fully meet PLOS ONE’s publication criteria as it currently stands. Therefore, we invite you to submit a revised version of the manuscript that addresses the points raised during the review process.

We look forward to receiving your revised manuscript.

Kind regards,

Xingwei Li, Ph.D.

Academic Editor

PLOS ONE

Journal Requirements:

Additional Editor Comments:

All reviewers have pointed out significant flaws in this manuscript. I recommend that the authors carefully revise this manuscript in accordance with these suggestions.

Reviewers' comments:

Reviewer #1: This manuscript has high innovative value in method, but the language expression is more obscure. It is suggested that the authors should revise the language and grammar problems, and this manuscript should be reviewed again.

Reviewer #2: The authors are recommended to construct a framework of MCDM methods and show where TOPSIS is situated in this framework. Instead of categorising the methods based on where they were originated (c.f. the 2nd paragraph of Introduction), the authors are recommended to classify them according to the methodology (e.g. parametric/non-parametric; pairwise/threshold, etc.; c.f. Diaz-Balteiro et al., 2017).

Diaz-Balteiro, L., Gonzalez-Pachon, J., & Romero, C. (2017). Measuring systems sustainability with multi-criteria methods: A critical review. European Journal of Operational Research, 258(2), 607-616. doi:10.1016/j.ejor.2016.08.075

The authors mention a considerable number of MCDM methods in Introduction but explain little about the applicability and feasibility of each method. A recommendation is that authors answer the following questions to justify the selection of TOPSIS: Which scenario(s) is each method to be used for? What is the context and features of the problem you are to solve in this paper (i.e. the sample case)?

In Equation (4), why PIS and NIS are function of weight (w) rather than weighted normalised value (v)?

As is shown in Equation (5), Euclidean distance is computed and thereby the correlation between criteria is uncaptured. The authors are recommended to discuss on this.

Please define the problem (L.195 – L.293) in a concise manner and focus the pointers relevant to the applicability and feasibility of TOPSIS.

The authors are recommended to state the contribution of this paper in an explicit manner. The findings of this paper are explicit (c.f. Conclusion), but what are the contributions of these findings? Are they relevant to the long-standing criticism of TOPSIS like rank reversal (Shin et al., 2013)?

Please show the author’s name and year of publication in the in-text citation to improve the readability.

Please add captions to the graphics.

Proofreading is highly recommended.

---

## [Author Response · Author response to Decision Letter 0]

20 May 2022

Reviewer #1

C1: This manuscript has high innovative value in method, but the language expression is more obscure. It is suggested that the authors should revise the language and grammar problems, and this manuscript should be reviewed again.

R: Thank you very much. The manuscript has undergone extensive and professional language proofreading, which has been carried out by an external entity.

Reviewer #2

C1: The authors are recommended to construct a framework of MCDM methods and show where TOPSIS is situated in this framework. Instead of categorising the methods based on where they were originated (c.f. the 2nd paragraph of Introduction), the authors are recommended to classify them according to the methodology (e.g.parametric/non-parametric; pairwise/threshold, etc.; c.f. Diaz-Balteiro et al., 2017). 

R: The framework of MCDM methods is created and added in the Introduction. The recommended source is also added in the references.

C2: The authors mention a considerable number of MCDM methods in Introduction but explain little about the applicability and feasibility of each method. A recommendation is that authors answer the following questions to justify the selection of TOPSIS: Which scenario(s) is each method to be used for? What is the context and features of the problem you are to solve in this paper (i.e. the sample case)?

R: The applicability and feasibility of each method discussed in the Introduction is described via 2 new classifications, this way the selection of TOPSIS is explained. The sample case is described too. 

C3: In Equation (4), why PIS and NIS are function of weight (w) rather than weighted normalised value (v)?

R: Thanks for reporting an error in the formula. The error is fixed.

C4: As is shown in Equation (5), Euclidean distance is computed and thereby the correlation between criteria is uncaptured. The authors are recommended to discuss on this.

R: Euclidean distance is the square root of the sum of squared differences between corresponding elements of the two vectors. It's a reverse measure of similarity: the smaller the Euclidean distance, the greater the similarity between X and Y. Euclidean distance is appropriate for data measured on the same scale that is our case (see criteria R1-R6). Correlation coefficient is (inversely) related to the Euclidean distance when the variables have been standardized, removing differences in scale.

C5: Please define the problem (L.195 – L.293) in a concise manner and focus the pointers relevant to the applicability and feasibility of TOPSIS.

R: The problem is defined in the context of TOPSIS and its applicability and feasibility.

C6: The authors are recommended to state the contribution of this paper in an explicit manner. The findings of this paper are explicit (c.f. Conclusion), but what are the contributions of these findings? Are they relevant to the long-standing criticism of TOPSIS like rank reversal (Shin et al., 2013)?

R: The contribution of this manuscript is added/defined in an explicit manner. 

C7: Please show the author’s name and year of publication in the in-text citation to improve the readability.

R: We totally agree with you that the author’s name and year of publication in the in-text citation improve the readability. The manuscript is prepared according to the formal requirements of the journal, which requires the authors to be listed using numbers.

C8: Please add captions to the graphics.

R: Captions are added to each figure. 

C9: Proofreading is highly recommended.

R: The manuscript has undergone extensive and professional language proofreading, which has been carried out by an external entity.

---

## [Decision Letter · Decision Letter 1]

4 Jul 2022

PONE-D-21-35126R1Similarity of TOPSIS Results based on Criterion Variability: Case Study on Public EconomicPLOS ONE

Dear Dr. Vavrek,

Thank you for submitting your manuscript to PLOS ONE. After careful consideration, we feel that it has merit but does not fully meet PLOS ONE’s publication criteria as it currently stands. Therefore, we invite you to submit a revised version of the manuscript that addresses the points raised during the review process.

We look forward to receiving your revised manuscript.

Kind regards,

Xingwei Li, Ph.D.

Academic Editor

PLOS ONE

Additional Editor Comments:

The authors put the detailed taxonomy of MCDM methods in the Introduction, which makes the Introduction less focused. For instead, the authors are recommended to elaborate on the taxonomy (some charts will improve the readability) in a following section and to justify the feasibility of TOPSIS thereby. You might go through the taxonomy (only one metric for classification but do not put origin and methods together) in Introduction in a succinct way but focus on the main issue and contribution of this manuscript in the Introduction.

Reviewers' comments:

**Comments to the Author**

Reviewer #1: The authors have done a good job of revision, and the paper meets the publication requirements of this journal.

Reviewer #2: The authors made significant revisions, which improves the readability of the manuscript. However, the manuscript needs better organisation.

The authors put the detailed taxonomy of MCDM methods in the Introduction, which makes the Introduction less focused. For instead, the authors are recommended to elaborate on the taxonomy (some charts will improve the readability) in a following section and to justify the feasibility of TOPSIS thereby. You might go through the taxonomy (only one metric for classification but do not put origin and methods together) in Introduction in a succinct way but focus on the main issue and contribution of this manuscript in the Introduction.

---

## [Author Response · Author response to Decision Letter 1]

9 Jul 2022

Reviewer #1

C1: The authors have done a good job of revision, and the paper meets the publication requirements of this journal.

R: Thank you very much for appreciating the adjustments made and the time devoted to it.

Reviewer #2

C1: The authors made significant revisions, which improves the readability of the manuscript. However, the manuscript needs better organisation. The authors put the detailed taxonomy of MCDM methods in the Introduction, which makes the Introduction less focused. For instead, the authors are recommended to elaborate on the taxonomy (some charts will improve the readability) in a following section and to justify the feasibility of TOPSIS thereby. You might go through the taxonomy (only one metric for classification but do not put origin and methods together) in Introduction in a succinct way but focus on the main issue and contribution of this manuscript in the Introduction.

R: Thank you very much for appreciating the adjustments made. The Introduction is rewritten. Origin description is deleted, only one metric for classification is described using a chart. With these changes (also text reducing), more emphasis is placed on the feasibility of TOPSIS and we more focus on the main issue and contribution of our manuscript.

---

## [Editor Report · Decision Letter 2]

12 Jul 2022

Similarity of TOPSIS Results based on Criterion Variability: Case Study on Public Economic

PONE-D-21-35126R2

Dear Dr. Vavrek,

We’re pleased to inform you that your manuscript has been judged scientifically suitable for publication and will be formally accepted for publication once it meets all outstanding technical requirements.

Kind regards,

Xingwei Li, Ph.D.

Academic Editor

PLOS ONE

---

## [Editor Report · Acceptance letter]

15 Jul 2022

PONE-D-21-35126R2 

Similarity of TOPSIS Results based on Criterion Variability: Case Study on Public Economic 

Dear Dr. Vavrek:

I'm pleased to inform you that your manuscript has been deemed suitable for publication in PLOS ONE. Congratulations! Your manuscript is now with our production department. 

Kind regards, 

on behalf of

Prof. Dr. Xingwei Li 

Academic Editor

PLOS ONE